# Comparative Analysis of Granular and Membrane Filters for Rainwater Treatment

**Celimar Azambuja Teixeira [1],\* and Enedir Ghisi [2]**

1   Department of Civil Engineering, Federal Technological University of Paraná, Curitiba, PR 81280-340, Brazil
2   Laboratory of Energy Efficiency in Buildings, Department of Civil Engineering, Federal University of Santa Catarina, Florianópolis, SC 88040-900, Brazil; enedir.ghisi@ufsc.br
\*   Correspondence: celimar@utfpr.edu.br; Tel.: +55-41-999585622

**Abstract:** The objective of this study was to compare the efficiency of rainwater treatment using two types of filters: one with filtration materials (gravel, sand, and anthracite) and the other employing membranes. In both cases, the quality of the rainwater after passing through the filter met the standards required by NBR 15527:2007 (Brazilian Association of Technical Standards (ABNT)) and the United States Environmental Protection Agency (EPA) for non-potable uses according to the parameters pH, temperature, turbidity, ammonia, nitrate, nitrite, alkalinity, and calcium hardness. The results obtained were also compared with Directive 2914/2011 of the Brazilian Ministry of Health, which deals with water potability, and with Resolution 357/2005 of CONAMA (Brazilian National Council for the Environment), which is applied to surface water bodies, especially rivers, and establishes the possibility of direct contact of the water with human skin. It was concluded that the rainwater obtained from both filters could be used for non-potable uses, such as toilet flushing, garden irrigation, and sidewalk cleaning, as well as for direct-contact activities, such as bathing and washing clothes.

**Keywords:** water conservation; rainwater use; rainwater treatment; filters

## 1. Introduction

Scarcity in water supply is related to the availability and quality of water sources, along with an increase in consumption due to urban development, human activities, and industrialization.

Due to a worsening of the quality of water sources and scarcity, alternative sources and ways to promote the rational use of water are required in order to guarantee a stable water supply [1]. In this context, rainwater appears to be a potential alternative source, and several researchers consider it as one of the solutions for the crisis in potable water supply [2].

Clean water and sanitation for all is the Sustainable Development Goal 6, one of 17 Sustainable Development Goals set by the United Nations in 2015. However, increasing level of water stress indicates a substantial use of water resources, which has great impact on the sustainability of these resources and increases the potential for conflict among users. More than 2 billion people live in countries experiencing water stress. Recent estimates show that 31 countries experience water stress between 25% (which is defined as the minimum level of water stress) and 70%. An additional 22 countries are above the 70% level and therefore in a severe state of water stress. Water use has increased worldwide by about 1% per year since the 1980s, and the world demand for water is expected to continue increasing at a similar rate until 2050, meaning an increase of between 20% and 30% relative to the current level of water use [3].

Rainwater harvesting and use has become a common practice, mainly in arid or remote zones where the supply of water through distribution networks is either not profitable or not technically

viable [1]. Thus, rainwater is a potential alternative source of water for human supply for potable and non-potable uses [4].

Rainwater harvesting not only provides a water source to increase the supply but can also play an important role in the involvement of communities in water management. This practice should therefore be encouraged and promoted by governments. The external factors and the characteristics of water collection surfaces as well as the impact of the cleanliness and age of the equipment, cisterns, pipes, and gutters affect the quality of the water collected [5].

Rainwater collected from roofs can be contaminated by microbial pathogens originated from fecal contamination by birds and small animals that have access to the roof areas and the rainwater tanks [2].

Another important factor is the type of roof material because this can affect the physicochemical quality of the rainwater. The effect of the roof material (asphalt, glass fiber, metal, and ceramic roofing) on the quality of the rainwater collected has previously been examined, and the results obtained showed the need to treat the rainwater collected from all types of roofs [6].

Another important factor regarding the quality of rainwater collected from rooftops is that water obtained from the first few moments of rainfall should be discarded because the initial runoff from a rooftop may contain pollutants in relatively higher concentrations. Thus, a first flush diverter, also known as a roof washer, must be installed so that the first volume of rainwater is directed away from the rainwater catchment system [7].

The filtration process consists of a combination of physical, chemical, and, in some cases, biological processes, which enable the removal of suspended and colloidal particles as well as microorganisms present in the water that flows through a porous medium, also called a filtration medium.

For single-layer water filters, a thickness of 0.55 m for the filtration material layer (sand) is recommended as well as an effective size of 0.4 to 0.5 mm, a coefficient of uniformity lower than 1.6, and grain size of 0.35 mm minimum and 1.2 mm maximum. For double-layer and treble-layer water filters, the thickness of the sand layer should vary from 15 to 30 cm, while the anthracite layer should be 45 to 60 cm. In upward-flow water filters, the minimum thickness required for the support layer is 0.40 m [8].

The support layer should be composed of rolled pebbles found in riverbeds, with sizes varying from 2 to 50 mm and a specific mass of 2.50 $g/cm^3$ [9].

Sand water filters are effective for the retention of solid matter in suspension, such as algae, other organic matter, and fine sand and silt particles. The better ability of a sand filter to remove organic matter when compared to other types of filters is because contaminants are collected as the water passes through the sand layer [10].

The removal of pesticides and nitrate from contaminated water via a post-treatment through filtration in a sand filter was studied by Aslan [11], who observed a significant removal of nitrates. The sand filter system improved the water quality considerably, providing good efficiency in the removal of turbidity and suspended solids. A significant removal of suspended solids from 60% to 68% was verified.

A membrane is a filter that eliminates compounds that are above the molecular weight cut-off point. The most appropriate membrane process to obtain water of good quality is dependent on the compounds to be eliminated from the water. The employment of membrane filtration is divided into four categories according to their cut-off points: microfiltration, ultrafiltration, nanofiltration, and reverse osmosis. The first two processes require operation below a pressure of 5 bar (equivalent to 500 KPa in the international system) and are mainly recommended for solid/liquid separation and the elimination of particles. Ultrafiltration retains mineral and organic particles as well as biological particles (algae, bacteria, and viruses), in addition to eliminating dissolved organic molecules [8].

A pre-filtration device to remove pathogens was developed by Zhang et al. [12]. Their results demonstrated that the combination of sand filtration and activated carbon technology with the chemical function of the pre-filtration device can be effective in removing viruses, bacteria, protozoans, and algae, besides removing water turbidity and odor.

Turbidity is commonly used to represent the presence of particles in water, which is an important parameter for water quality. These particles can be inorganic solids (silt, sand, or clay) or organic matter

(algae, bacteria, etc.). In the case of rainwater, the organic matter can originate from mosses, lichens, bird feces, or twigs and leaves that are present on the collection surface. In some cases, suspended matter from the atmosphere can be present [1].

The development of a filtration system with sand and activated carbon for rainwater treatment in Malaysia was investigated by Shaheed et al. [13]. The activated carbon was obtained from coconut shell of local origin, and it was activated with the use of readily available salt instead of a high-technology procedure that would require a chemical reagent. The system produced effluents that met the potable water standards for the parameters pH, dissolved oxygen (DO), biochemical oxygen demand (BOD$_5$), chemical oxygen demand (COD), total suspended solids (TSS), and ammoniacal nitrogen (NH$_3$-N), in addition to decreasing the *Escherichia coli* (*E. coli*) population.

Yan et al. [14] presented a device for filtration treatment with activated carbon and disinfection by ozonation to obtain potable water from rainwater in England. The field tests showed that the device was capable of reaching this objective with a reduction in the energy consumption as the system uses solar and oleic energy sources and does not consume energy from the local grid.

A filtration process integrating a gravity-driven membrane (GDM) with granular activated carbon (GAC) was evaluated by Tang et al. [15]. The system operated for 193 days, and the presence of granular activated carbon resulted in a much higher stable flow (6 L m$^{-2}$ h$^{-1}$ compared with 2 L m$^{-2}$ h$^{-1}$ for the system using only membranes). Furthermore, the combination of GAC with GDM provided a significant removal of dissolved organic compounds (50% to 70%) due to adsorption and biodegradation processes.

In view of the above, the objective of this study was to compare the efficiency of rainwater treatment using two types of filters: a filter composed of gravel, sand, and activated carbon and a membrane filter using needle-punched nonwoven polyester geotextile membrane.

## 2. Materials and Methods

### 2.1. Place of Study

Two types of filters were developed for rainwater treatment, and they were comparatively evaluated. The first was a downward flow sand filter and the second was a membrane filter.

The filters were installed in the city of São José, in the state of Santa Catarina, southern Brazil, next to each other to ensure that they received the same rainwater with the same characteristics of quality and intensity.

The building where the two filters were installed for rainwater treatment is located at a distance of around 100 m from highway BR 277, which receives a heavy flow of traffic and lies at approximately 200 m from the sea. The installation site is shown in Figure 1.

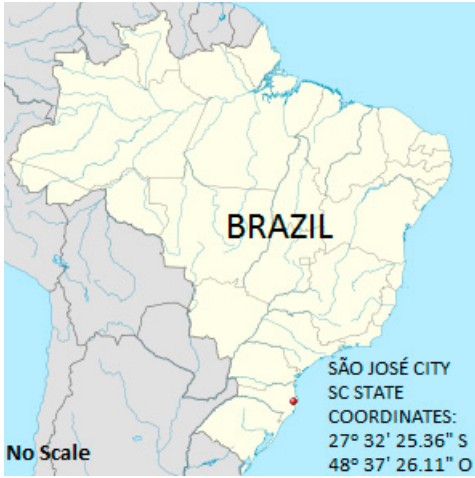

**Figure 1.** Location of site for installation of filters for rainwater treatment.

## 2.2. Description of Sand Filter

The catchment area was 50.70 m$^2$ and composed of fibrocement roof tiles, gutters, pipes, a first flush device, and a reservoir. The system works by gravity in the following sequence. The water flows over the roof and is captured by the gutter. Before treatment, the water runs through a screen to separate any coarse material (twigs and leaves) that may be on the roof tiles. The water is then conducted through a piping system to the first flush device, which discards the first two millimeters of rainwater. When the first flush device is full, the water is forced to flow through the filter by means of a T-connection. After filtration, the water passes to a reservoir composed of a 310-liter water tank.

The first flush device should store the first two millimeters of rainfall. In this study, a 2-mm hole was made at the end of the pipe of the first flush device so that it could empty automatically. However, it was noted that this 2-mm hole did not work properly due to clogging caused by the dirt in the atmosphere and on the roof, which was brought by the rainwater. Thus, cleaning was required at the end of each rain event. Details of the sand filter are shown in Figure 2.

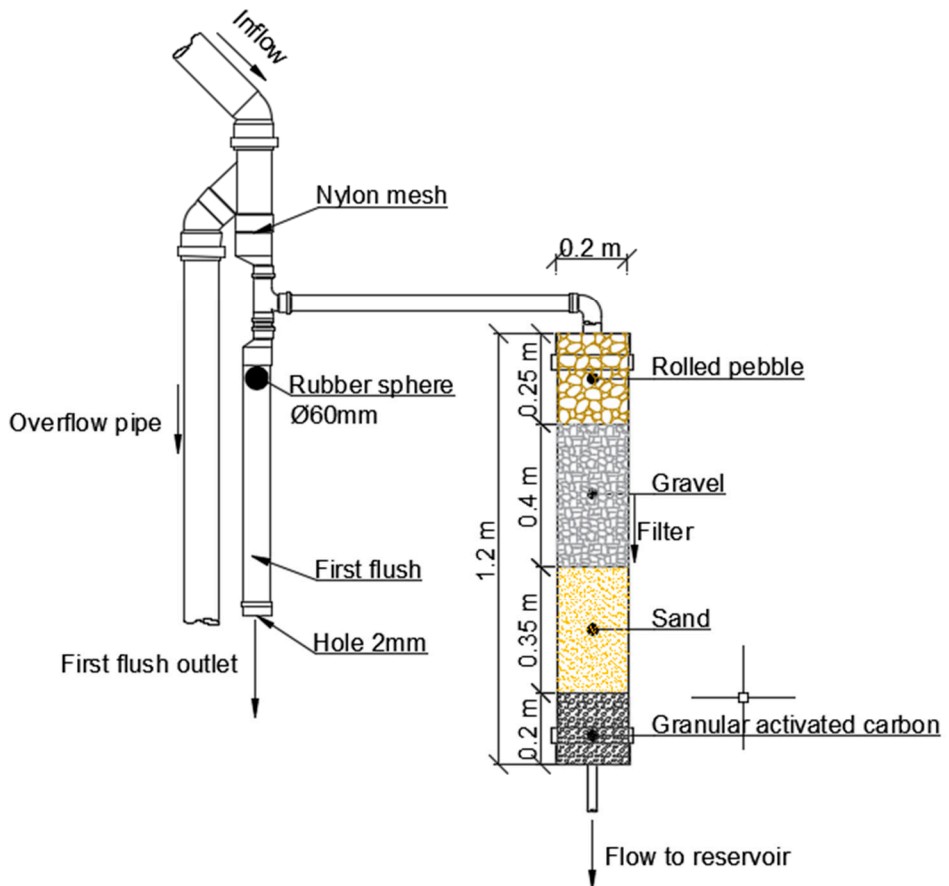

**Figure 2.** Schematic showing the sand filter.

The downward flow filter was made of polyvinyl chloride (PVC) and was composed of a support layer of 0.25 m, containing rolled pebbles with a diameter between 2.5 and 3.8 cm, and a filtration layer that was composed of 0.40 m of gravel, 0.35 of sand, and 0.20 m of GAC (Figure 3).

Sand, gravel, and rolled pebbles were washed with water and dried in an oven (Lucadema N 1040) at 100 °C for a period of 24 h before filter construction to reduce initial turbidity and for the filter to reach maturation faster. The activated carbon was regenerated by means of a thermal regeneration process; it was activated in an oven at 300 °C for 24 h.

In order to characterize the filtration materials, tests were carried out to determine the granulometry; void index; specific mass; pH; volatile matter content; ash content; moisture content and bulk density of

the sand, gravel, and activated carbon; and the iodine number of the activated carbon. All granulometry tests were carried out in the Soil Mechanics Laboratory at UTFPR, Campus Ecoville. Table 1 shows the parameters and the reference methodologies used to characterize the filter materials.

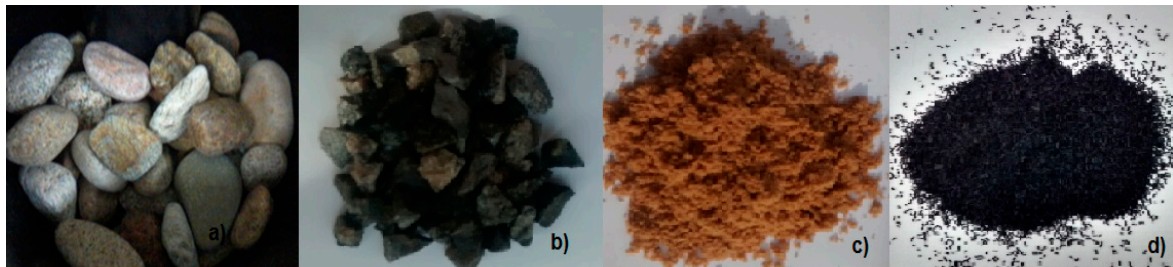

**Figure 3.** Materials used in the filter: (**a**) rolled pebbles; (**b**) gravel; (**c**) sand; (**d**) granular activated carbon (GAC).

**Table 1.** Parameters and the reference methodologies used to characterize the filter materials.

| Parameter | Sand | Gravel | GAC | Methodology | Unit |
|---|---|---|---|---|---|
| Granulometry | X | X | X | ABNT NBR NM 248 (2001) [1] | - |
| pH | X | X | X | ASTM D 3838-80 (1999) [2] | - |
| Moisture content | X | X | X | ASTM D 2867-04 (2004) [3] | % |
| Volatile matter content | X | X | X | ASTM D 5832-98 (2003) [4] | % |
| Ash content | X | X | X | ASTM D 2866-94 (1999) [5] | % |
| Specific mass | X | X | X | ABNT NBR NM 23 (2001) [6] | g/cm$^3$ |
| Bulk density | X | X | X | ABNT NBR 52 (2009) [7] Sand ABNT NBR 53 (2009) [8] Gravel ABNT NBR 12076 (1991) [9] CAG | g/cm$^3$ |
| Iodine number | | | X | ABNT NBR 12073 (1991) [10] | mg/g |
| Void index | X | X | X | ABNT NBR 45 (2006) [11] | % |

Note: [1]ABNT: Brazilian Association of Technical Standards. NBR NM 248:2001 [16]; [2]ASTM: American Society of Testing and Materials D 3838-80:1999 [17]; [3] American Society of Testing and Materials D 2867-04:2004 [18]; [4]American Society of Testing and Materials D 5832-98:2003 [19]; [5]American Society of Testing and Materials D 2866-94:1999 [20]; [6]ABNT: Brazilian Association of Technical Standards. NBR NM 23:2001 [21]; [7]ABNT: Brazilian Association of Technical Standards. NBR 52:2009 [22]; [8]ABNT: Brazilian Association of Technical Standards. NBR 53:2009 [23]; [9]ABNT: Brazilian Association of Technical Standards. NBR 12076:1991 [24]; [10]ABNT: Brazilian Association of Technical Standards. NBR 12073:1991 [25]; [11]ABNT: Brazilian Association of Technical Standards. NBR 45:2006 [26].

A summary of the results for the characterization of the filtration materials (sand, gravel, and granular activated carbon) is shown in Table 2.

**Table 2.** Average values and standard deviation of the parameters used for the physicochemical characterization of the sand, gravel, and activated carbon.

| Parameter | Sand | Gravel | Activated Carbon |
|---|---|---|---|
| pH | 6.8 ± 0.1 | 8.8 ± 0.1 | 6.7 ± 0.1 |
| Volatile matter content (%) | 1.0 ± 0.1 | 2.9 ± 0.1 | 50.8 ± 0.1 |
| Moisture content (%) | 2.38 ± 0.01 | 0.05 ± 0.01 | 48.73 ± 0.01 |
| Ash content (%) | 1.11 ± 0.01 | 6.96 ± 0.01 | 6.25 ± 0.01 |
| Specific mass (g/cm$^3$) | 2.61 ± 0.01 | 2.69 ± 0.01 | 1.27 ± 0.01 |
| Bulk density (g/cm$^3$) | 1.47 ± 0.02 | 1.38 ± 0.02 | 0.63 ± 0.02 |
| Void Index (%) | 43.8 ± 0.1 | 48.8 ± 0.1 | 32.9 ± 0.1 |
| Iodine number (mg/g) | - | - | 665.86 ± 0.01 |

The filtration materials used in the filter had degrees of uniformity of 4.9 for sand and 1.9 for gravel, which classifies them as highly uniform.

In Table 3, the granulometric data for the sand and gravel, obtained through granulometry tests, are provided.

**Table 3.** Granulometric data for sand and gravel.

| Material | Minimum Diameter (mm) | Maximum Diameter (mm) | $D_{10}$ (mm) | $D_{60}$ (mm) | $\frac{D_{60}}{D_{10}}$ (Coefficient of Uniformity) |
|---|---|---|---|---|---|
| Sand | 0.15 | 4.76 | 0.27 | 1.32 | 4.90 |
| Gravel | 4.76 | 19.10 | 7.49 | 14.60 | 1.90 |

### 2.3. Description of Membrane Filter

In this study, the efficiency of the membrane filter system was evaluated. Membranes of needle-punched nonwoven polyester geotextile (Bidim RT31) were used in the rainwater harvesting treatment process. The membrane used had the following hydraulic properties: permeability normal to plane, kn = 0.37 cm/s; permittivity, $\psi$ = 0.8 s$^{-1}$; apparent opening size, O95 = 0.125mm.

The model evaluated was based on that proposed by Vieira et al. [27]. In Figure 4, the operation stages and the main materials used in the making of the filter are described.

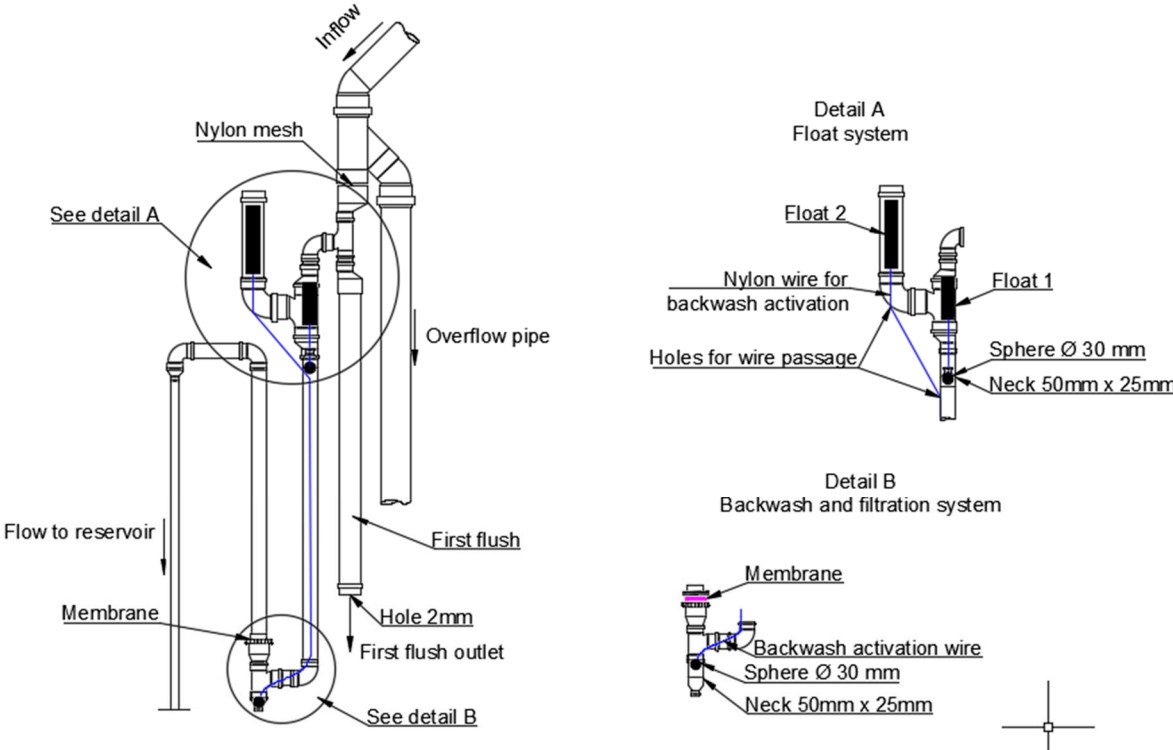

**Figure 4.** Schematic showing the assembly of the membrane filter system.

The membrane filter was designed to carry out the backwashing process automatically and mechanically, without requiring manual operation. The system works as follows. The water that flows through the gutter fills up the first flush device and then enters the treatment system. The first pipe is filled with water and has a downward flow direction, while the second pipe has an upward flow direction and the water passes through the filtration membrane. After filtration, the water flows to the reservoir, which is totally sealed and does not allow water to leak. In this case, a water storage tank of 310 L with a threaded lid was used.

When the whole system is filled with water, float 1 rises inside the pipe and a rubber sphere with a diameter equal to 30 mm (sphere 1) interrupts the inward flow of the filtration system. At this moment, the rainwater is directed to another set of pipes containing float 2, which is then set in motion and rises within the pipe.

Float 2 is connected, through a steel cable, to a second rubber sphere with a diameter equal to 30 mm (sphere 2), which is placed at the connection below the filtration membrane. Thus, when float 2 rises within the pipe, the steel cable pulls sphere 2. The water flow in the pipe is thereby released and carries out the backwashing of the filtration membrane.

The two floats were built from pieces of PVC pipes with a diameter equal to 50 mm, closed at the ends with caps with the same diameter.

## 2.4. Qualitative Parameters

For comparison purposes, physicochemical parameters were selected to provide representative data on the contaminants from which a relation between the trends observed during the period of study could be established in order to evaluate the efficiency of the rainwater treatments applied using the sand and membrane filters.

The parameters determined were pH, temperature, turbidity, alkalinity, and calcium hardness, along with ammonia, nitrite, nitrate, and phosphate concentrations. These parameters were used to compare the efficiency of treatment of the filters and compose a database on the quality of rainwater. In addition, the limits of each one according to standards and guidelines for water quality were assessed.

Although the measurement of water temperature is controlled by external environmental conditions, this parameter especially affects pH and was therefore considered. The parameters temperature, alkalinity, and calcium hardness were considered, even though they are not specified in any Brazilian resolution or standard.

The pH, temperature, and turbidity were determined based on the methodology established by the Standard Methods for Examination of Water and Wastewater [28]. The ammonia, nitrite, nitrate, and phosphate contents and the alkalinity and calcium hardness were determined in a multiparameter bench photometer (Hanna HI 83099). The methods for the determination of each parameter are described in Table 4.

**Table 4.** Methodology and precision for the determination of parameters using a Hanna HI 83099 photometer.

| Parameter | Methodology | Precision |
|---|---|---|
| Ammonia | Nesseler, adapted from ASTM Manual of Water and Environmental Technology, D-1426-92 | ±0.04 mg/L ± 4% for reading at 25 °C |
| Nitrite | Adapted from US EPA 354.1 (diazotization method) | ±0.06 mg/L ± 4% for reading at 25 °C |
| Nitrate | Adapted from the reduction of cadmium | ±0.5 mg/L ± 10% for reading at 25 °C |
| Alkalinity | Colorimetric method | ±5 mg/L ± 10% for reading at 25 °C |
| Calcium Hardness | Adapted from calmagite method | ±0.11 mg/L ±5% for reading at 25 °C |

The sample collection was arranged so that the analysis could be carried out approximately twice per month, the period being dependent on the occurrence of rain. In total, 15 collections were performed between December 2015 and August 2016.

The water samples were collected from the rainwater storage reservoirs installed after each of the filters. Samples were taken from approximately 10 cm below the water surface to avoid disturbing the sediments at the bottom. Samples were also collected from a recipient that stored the precipitated rain, that is, the untreated rainwater. The samples collected were immediately stored in previously sterilized containers and maintained in ice prior to analysis.

The experimental data obtained from the physicochemical determinations are reported in box-plot graphs, which allow the main trends and the variability of the sampling data to be observed. In these graphs, the median (50th percentile), the lower (25th percentile), and the upper (75th percentile) quarters as well as measures of data dispersion, such as the minimum and maximum values, are included.

The results obtained were compared with the values established in Resolution 357/2005 of the Brazilian National Council for the Environment [29], which provides classifications for water bodies and environmental guidelines for their inclusion as well as establishing standards for effluent discharge. They were also compared with the values defined by the United States Environmental Protection Agency (EPA) [30], which sets the guidelines for water reuse and rainwater use.

The results were also compared with MS Directive 2914/2011 of the Brazilian Ministry of Health [31], which recommends procedures for the control and monitoring of the quality of water for human consumption and potability standards, and with the values given in NBR 15527:2007 (Brazilian Association of Technical Standards) [32], which establishes the requirements for non-potable use of rainwater collected from roofs in urban areas.

According to NBR 15527:2007 [32], rainwater must go through a disinfection process, which may be chlorination, ultraviolet rays, ozone, or other processes, defined at the discretion of the designer. The parameters measured in this research are sufficient to guarantee non-potable use, considering that rainwater must go through a disinfection process after filtration.

A comparison of the rainwater quality with the values given in CONAMA Resolution 357/2005 [29] was also carried out in order to establish which treatment is suitable for obtaining water for potable use.

## 3. Results and Discussion

### 3.1. Rainfall Analysis

The daily rainfall was evaluated from 01/12/2015 to 07/08/2016, and a lack of rain was verified in 130 days during this study period. The maximum precipitation was 93.4 mm on 03/03/2016. Figure 5 shows the daily rainfall behavior for the city of São José, highlighting the average daily rainfall and also considering the days without rain. These data were obtained from the National Institute of Meteorology (INMET) [33].

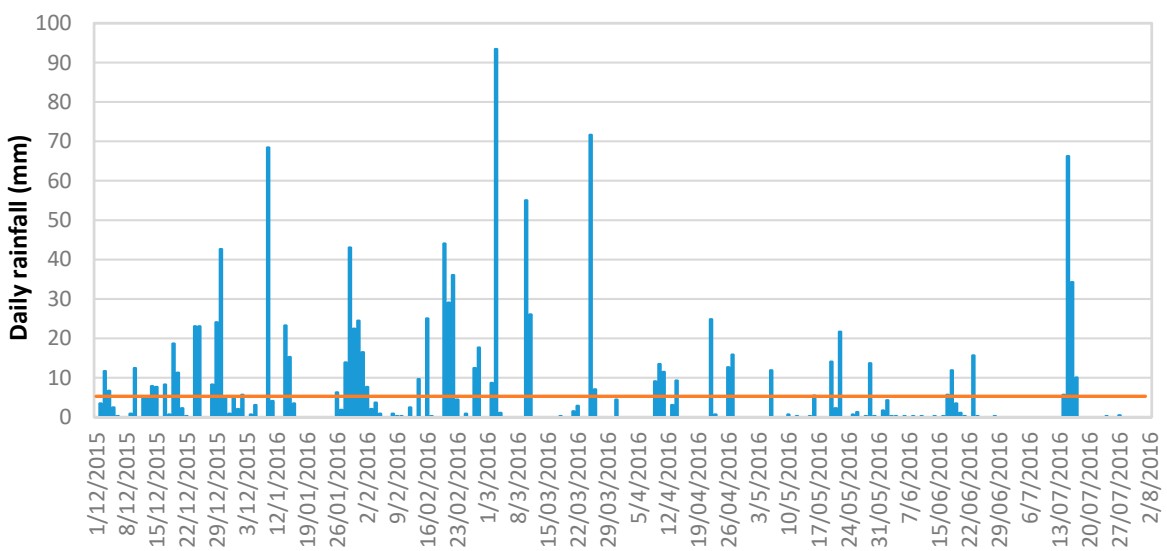

**Figure 5.** Daily rainfall in the city of São José from 01/12/2015 to 07/08/2016.

### 3.2. Physicochemical Analysis of Rainwater

The incorporation of impurities present on the collection surface into the rainwater occurs due to the direct contact of rainwater with the roof.

As the building where the two filters for the rainwater harvesting treatment were installed is located approximately 100 m from a highway with a high traffic load, it was verified through the pH data that the rainwater was within the standard for normality in relation to acidity and was not considered as acid rain (i.e., pH < 5, which was not the case).

Box-plot graphs containing median, minimum, maximum, 1st and 3rd quarters, and interquartile range of the parameters evaluated are shown in Figure 6.

The filtration materials used in the sand filter had a degree of uniformity equal to 4.9 for the sand and 1.9 for the gravel, making them highly uniform. Sezerino [34] obtained coefficients of uniformity equal to 5.70 for sand and 1.89 for gravel in the characterization of filtration materials.

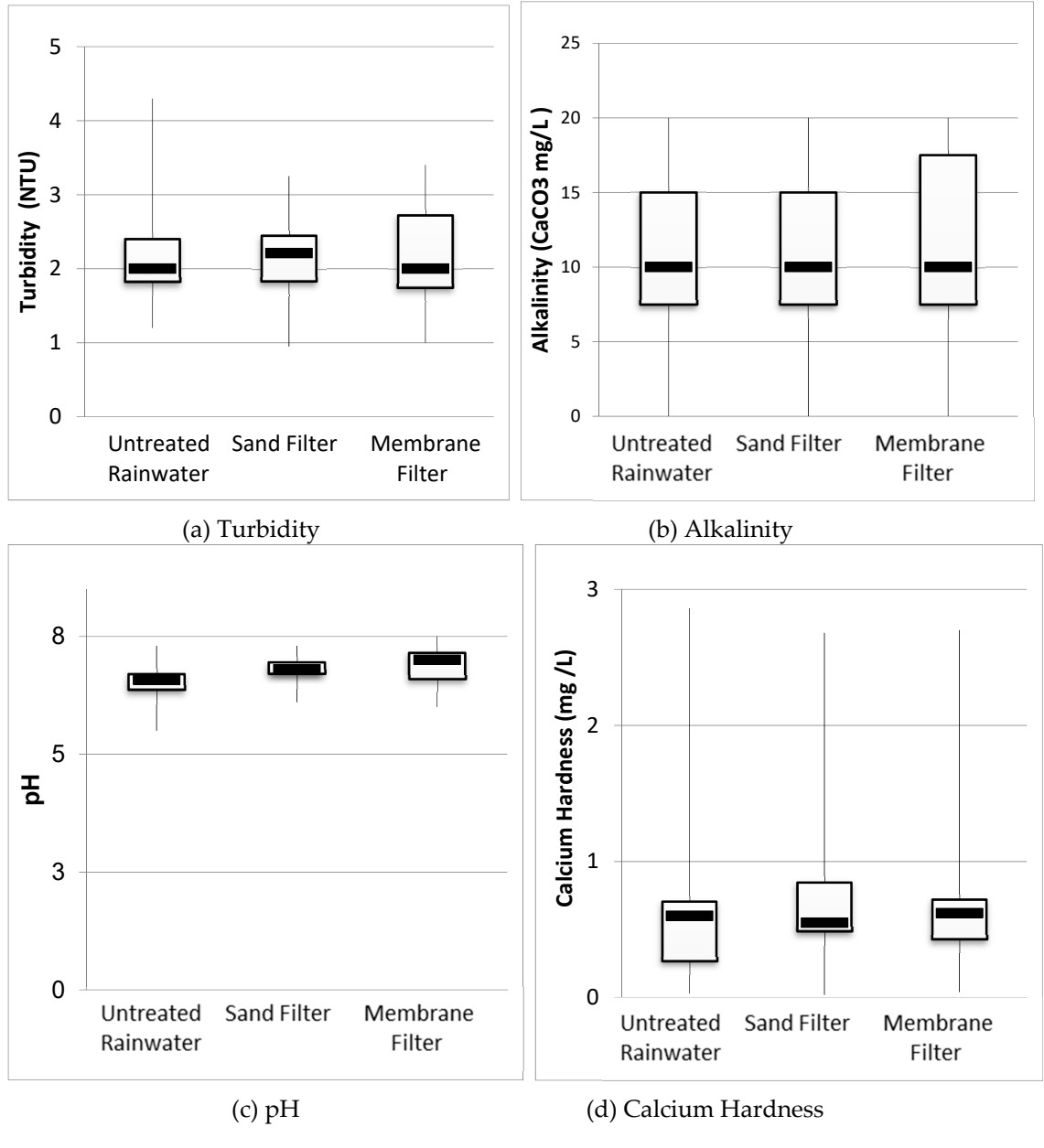

(a) Turbidity　　　　　　　　　　　　　　(b) Alkalinity

(c) pH　　　　　　　　　　　　　　(d) Calcium Hardness

**Figure 6.** *Cont.*

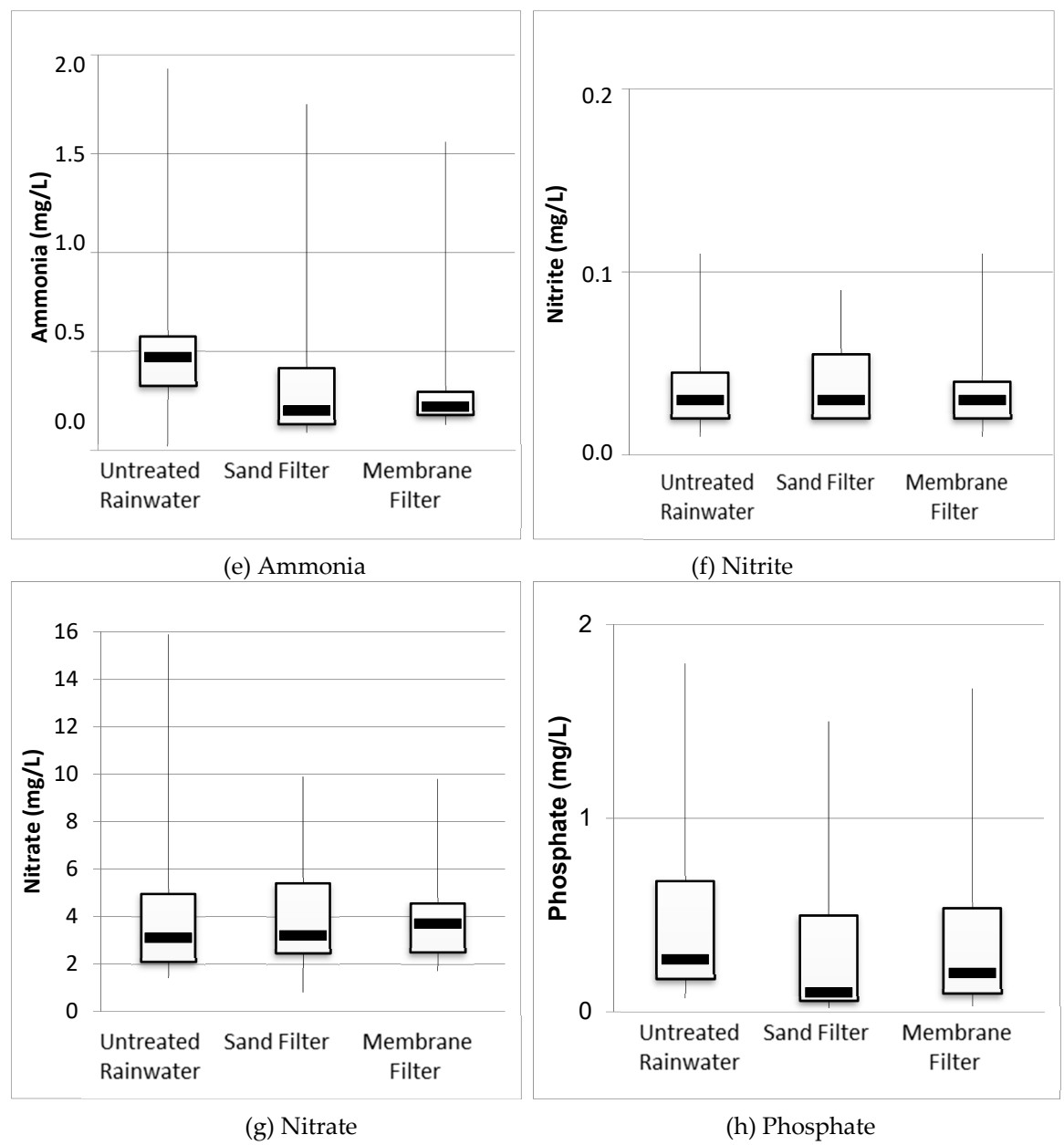

**Figure 6.** Box-plot graphs with median, minimum, maximum, 1st and 3rd quarters, and interquartile range of the parameters evaluated.

According to Sezerino [34], a coefficient of uniformity lower than or equal to 5 is recommended for sand. Coelho and Di Bernardo [8] obtained coefficients of uniformity equal to 2.0 and lower than 1.7 for sand and granular activated carbon, respectively. Brinck [9] obtained coefficients of uniformity equal to 1.76 and 1.97 for different sands and 1.30 and 1.96 for different anthracites. The lower the value for the coefficient of uniformity, the more uniform the granular material, the deeper the penetration of impurities, and the longer the filtration run time will be.

Table 5 provides a statistical summary of the results for the physicochemical analysis of the rainwater collected from the sampling points and allows a comparison with the values established in guidelines for water quality, in Brazil, that is, CONAMA Resolution 357/2005 [29], NBR 15527:2007 [32], and MS Directive 2914/2011 [31]. The results were also compared with the values given in the US EPA [30].

**Table 5.** Statistical results for the physicochemical analysis of rainwater collected at the sampling points.

| Parameters | N° | Untreated Rainwater | | | Sand Filter | | | Membrane Filter | | | Standards and Reference Values | |
|---|---|---|---|---|---|---|---|---|---|---|---|---|
| | | Max. | Min. | Average (SD) | Max. | Min. | Average (SD) | Max. | Min. | Average (SD) | | |
| pH | 15 | 7.3 | 5.5 | 6.5 (0.4) | 7.3 | 6.1 | 6.8 (0.3) | 7.5 | 6 | 6.9 (0.4) | US EPA[1] | 6.0–9.0 |
| | | | | | | | | | | | MS[2] | 6.0–9.5 |
| | | | | | | | | | | | CONAMA[3] | 6.0–9.0 |
| | | | | | | | | | | | NBR 15527[4] | 6.0–8.0 |
| Temperature (°C) | 15 | 29.0 | 15.8 | 22.9 (4.8) | 29.0 | 17.8 | 23.2 (4.6) | 29.6 | 18.5 | 24.2 (4.9) | No recommended value | |
| Alkalinity (mg/L) | 15 | 20.0 | 0.00 | 9.33 (5.44) | 20.00 | 0.00 | 11.67 (5.96) | 20.00 | 0.00 | 11.67 (6.24) | No recommended value | |
| Calcium Hardness (mg/L) | 15 | 2.86 | 0.03 | 0.63 (0.65) | 2.68 | 0.02 | 0.78 (0.66) | 2.7 | 0.04 | 0.7 (0.60) | No recommended value | |
| Turbidity (NTU) | 15 | 4.3 | 1.3 | 2.3 (0.9) | 3.2 | 0.9 | 2.0 (0.5) | 3.4 | 1 | 2.1 (0.8) | US EPA[1] | ≤2 NTU |
| | | | | | | | | | | | MS[2] | ≤0.5 NTU |
| | | | | | | | | | | | CONAMA[3] | ≤40 NTU |
| | | | | | | | | | | | NBR 15527[4] | <2.0 NTU* |
| NH₃ (mg/L) | 15 | 1.93 | 0.02 | 0.53 (0.42) | 1.75 | 0.09 | 0.35 (0.40) | 1.56 | 0.13 | 0.36 (0.38) | MS[2] | ≤1.5 mg/L |
| | | | | | | | | | | | CONAMA[3] | ≤2 mg/L |
| NO₂⁻ (mg/L) | 15 | 0.11 | 0.01 | 0.04 (0.03) | 0.09 | 0.02 | 0.04 (0.02) | 0.11 | 0.01 | 0.04 (0.03) | MS[2] | ≤1 mg/L |
| | | | | | | | | | | | CONAMA[3] | ≤1 mg/L |
| NO₃⁻ (mg/L) | 15 | 15.90 | 1.40 | 4.37 (3.68) | 9.90 | 0.80 | 3.97 (2.37) | 9.80 | 1.70 | 3.82 (1.99) | MS[2] | ≤10 mg/L |
| | | | | | | | | | | | CONAMA[3] | ≤10 mg/L |
| PO₄³⁻ (mg/L) | 15 | 1.80 | 0.07 | 0.48 (0.48) | 1.50 | 0.02 | 0.30 (0.35) | 1.67 | 0.03 | 0.35 (0.40) | No recommended value | |

Note: [1] US EPA: The United States Environmental Protection Agency. Guidelines for Water Reuse. EPA/600/R-12/618 [30]; [2] MS: Directive 2914/2011 of the Brazilian Ministry of Health, which deals with water potability [31]; [3] CONAMA: Brazilian National Council for the Environment. Resolution 357 on 17 March 2005 [29]; [4] ABNT: Brazilian Association of Technical Standards. NBR 15527:2007 [32]; * For less restrictive purposes, recommended turbidity is less than 15 in ABNT NBR 15527: 2007 [29].

It can be noted in Table 5 that the average and the standard deviation values for the pH of the rainwater samples collected post-treatment met the standards established by the abovementioned guidelines.

Regarding the ammonia and the nitrite concentrations, for all sampling points, the results remained within the values established by CONAMA Resolution 357/2005 [29] for a class 2 water body. Although this resolution is used to set limits for contaminants in a different type of water body, it was used herein as a standard for comparison purposes. However, the turbidity values obtained were higher than the values established by NBR 15527:2007 [32], mainly for the untreated rainwater. This could be due to the proximity of the building to highway BR 277.

Based on the turbidity results, both filters managed to retain particulate matter and therefore reduce the turbidity values. It was verified that the highest turbidity values for the untreated rainwater and the water after passing through the filters were obtained in the samples collected on 28/06/2016 and 13/07/2016. These results occurred after the longest period without rain, which allowed time for the accumulation of particulate matter in the atmosphere, thereby resulting in an increase in the turbidity of the rainwater.

Based on MS Directive 2914/2011 [31], which deals with water potability, the highest ammonia values were above the limit, while the nitrite values were acceptable.

In the case of nitrate, only the highest value obtained for the untreated rainwater was above the limit of MS Directive 2914/2011 [31] and CONAMA Resolution 357/2005 [29]. Both filters were efficient in the removal of ammonia but not in the removal of nitrite. Acceptable levels for ammonia, nitrite, and nitrate are not specified in the US EPA [30] or NBR 15527:2007 [32].

The average values obtained for the pH at all collection points met the limits of the guidelines used for comparison. In relation to the turbidity, only the samples collected after passing through the two filters met the limits of all guidelines. In the case of the untreated rainwater, the average turbidity values met the limits only of CONAMA Resolution 357/2005 [29].

The average ammonia, nitrite, and nitrate values were acceptable according to MS Directive 2914/2011 [31] and CONAMA Resolution 357/2005 [29]. These parameters are not addressed in the US EPA [30] and NBR 15527:2007 [32].

Honório et al. [35] carried out a study in the western region of Amazonia, Brazil, in order to evaluate the quality of untreated rainwater. Samples were collected in the towns of Parintins, Itapiranga, Tabatinga, Boa Vista, and Apuí and in the city of Manaus (where one sample was taken in an area covered by vegetation and another in an urban area). For the seven sampling points, average nitrate concentrations in the range of 4.7 to 32.0 mg/L were observed. The highest values were obtained in Manaus, which could be due to the rapid urban development and increase in the use of fossil fuels, given nitrite and nitrate concentrations in rainwater have been attributed to the combustion of fossil fuels [35].

Lee et al. [2] carried out a study on the quality of rainwater in the city of Gangneung, South Korea, considering three main collection points: untreated rainwater, rainwater after flowing off a roof, and rainwater from a reservoir. They obtained the following average values: pH 5.3 and nitrate 2.2 mg/L for rainwater flowing through a roof and pH 7.8 and nitrate 7.6 mg/L for rainwater accumulated in a reservoir.

In this study, the physicochemical parameters of the water in the first flush device were not evaluated. However, in both systems, a 2-mm hole made for their self-cleaning became clogged due to the accumulation of dirt inside the device, verifying the presence of impurities in the first flush water. Silva [36] also concluded that the first flush devices are more efficient at removing impurities than the filters.

For the sand filter, the removal efficiencies obtained were 13.0% for turbidity, 34.0% for ammoniacal nitrogen, and 10.0% for nitrate. In the case of the membrane filter, the corresponding values were 11.0%, 32.1%, and 13.6%, respectively.

For both filters, considering the average values, the treatment did not lead to significant changes in the pH and nitrite. In relation to pH, this is due to the fact that the values were close to the point of neutrality, and the filters did not reduce the nitrite levels because the concentrations in the untreated rainwater were low.

The sand and gravel filter did not require backwashing during the evaluation period as there was no change in the treatment efficiency. In relation to the membrane filter, the membrane was changed after a period of use of four months as it was worn and needed to be replaced in order to maintain the treatment efficiency.

On evaluating the results, it can be concluded that there were similar levels of efficiency in the rainwater treatment achieved with the sand and membrane filters. However, the sand filter can be considered more suitable for the rainwater treatment based on the maintenance criterion because it did not require any intervention during the period under study, maintaining the same treatment efficiency throughout the experiment.

The membrane filter, on the other hand, did require some maintenance, such as the adjustment of the membrane position and membrane replacement during the experimental period, to avoid a decrease in the treatment efficiency. Thus, considering the aspects of maintenance and treatment efficiency, the sand filter was found to be more suitable for rainwater treatment for the period evaluated in this study.

## 4. Conclusions

In this study, the efficiency of two filters for rainwater treatment was evaluated. As the rainwater in the evaluated region already has good quality, the results showed that the filters had similar efficiency between them, and the difference in efficiency was not significant. However, in regions of low rainwater quality, filters can be more efficient.

An important aspect in relation to the treatment of rainwater is the use of first flush devices to discard the initial few millimeters of rainfall. These devices are able to accumulate the impurities originating from the atmosphere, which are carried by the rain flowing over the roof. Their use represents one of the best ways to keep the rainwater filter systems clean and safe. In this study, the accumulation of impurities in the first flush devices was observed for both filters.

Thus, it is concluded that the rainwater treatment systems tested were efficient. This shows the potential of rainwater as an important source of water supply together with other sources, such as water utilities, water reuse, and possible underground water sources.

Based on the analysis carried out in this study, in areas with good quality of rainwater harvesting, the installation of first flush devices is indispensable, but whether a filtration system will be necessary should be evaluated based on the quality of the rainwater.

In addition, even with good-quality post-treatment rainwater, the use of such water for non-potable purposes must undergo a disinfection process for the safety of users.

**Author Contributions:** Conceptualization, C.A.T. and E.G.; methodology, C.A.T. and E.G.; software, C.A.T.; validation, C.A.T.; formal analysis, C.A.T. and E.G.; investigation, C.A.T.; resources, C.A.T. and E.G.; data curation, C.A.T.; writing—original draft preparation, C.A.T.; writing—review and editing, C.A.T and E.G.; visualization, C.A.T.; supervision, E.G.; project administration, C.A.T.; funding acquisition, C.A.T. and E.G.

**Funding:** This research was funded by Federal Technological University of Paraná and Federal University of Santa Catarina.

**Acknowledgments:** We appreciate the anonymous reviewers for their valuable comments and efforts to improve this manuscript.

**Conflicts of Interest:** The authors declare no conflict of interest.

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
