# Peer review of "Comparative Analysis of Granular and Membrane Filters for Rainwater Treatment"

_water, doi:10.3390/w11051004_

Round 1

Reviewer 1 Report

The manuscript deals with comparison of media and membrane filtration in rain water treatment targeting water reuse. The topic of rainwater harvesting and reuse is not new, but significant in the context of water scarcity, water reuse and nature-based solutions (e.g. see WWDR 2017, 2018). The overall opinion – this is rather a weak manuscript based on extremely limited experimental data without scientific value. It is observed that experimental design of this study was weak, while attention paid to review of various quality standards. It could have had a practical impact, but fragmentation of results prevents from publishing. Authors may consider converting this manuscript to a review paper.

After a brief introduction of the background that can be improved, authors provide limited information about technical characteristics of the membrane unit. Information about used materials is fragmented, and part of the filtration media characteristics is given in the results section. Experimental results of the study are very limited and scientific value is questionable.

In general, the manuscript does not seem to be novel and interesting. The impact and adds to the knowledge base are not clear.

Specific comments:

The title can be improved by pointing specific technologies that are compared.

Justification of selection of 8 parameters and if it is sufficient for concluding on non-potable reuse.

It is not clear why hardness has been considered, when it is known (and confirmed in Table 4) that rain water is soft.

Statements regarding scarcity can be strengthen with data/numbers on scarcity/droughts from SDGs.

Line 32: It is preferred to use functional/characteristic designation of membranes instead of trademark, while trademark and supplier can be specified in brackets.

Line 76-77: This statement is absolute for certain membrane types.

It would be good to wrap up the introduction with an objective statement, connecting literature findings with the research goal and selected methods.

Line 117: The statement regarding low costs in not supported in the place where it appears.

Figure 1: the city marker is in the sea.

Section 2.4: membrane characteristics are missing.

Tables 2, 3: Characteristics of filter media should be moved to Materials & Methods.

Figure 5: “in nature” is not a clear designation. Recommended: raw rainwater, after […] treatment. Later, in the Table 4 another designation is used “crude rainwater”. Authors must ensure consistence of terminology.

Figure 5: it is not clear if treatment effects are significant and selected treatment is needed, focusing on selected quality parameters.

Author Response

Reviewer #1

1) Reviewer 1 comment: The title can be improved by pointing specific technologies that are compared.

Authors Comment: The new title is “Comparative analysis between granular filter and membrane filter for rainwater treatment

2) Reviewer 1 comment:  Justification of selection of 8 parameters and if it is sufficient for concluding on non-potable reuse. It is not clear why hardness has been considered, when it is known (and confirmed in Table 4) that rain water is soft

Authors Comment: The explanation about this condition was inserted in text, between Lines 240 – 245, as follows:

The parameters determined were pH, temperature, turbidity, alkalinity and calcium hardness along with ammonia, nitrite, nitrate and phosphate concentrations. Although the measurement of water temperature is controlled by external environmental conditions, this parameter especially affects pH, so it was considered. The parameters temperature, alkalinity and calcium hardness, although not specified in any Brazilian resolution or standard, were used to compare the efficiency of the treatment and monitoring.

3) Reviewer 1 comment: Statements regarding scarcity can be strengthen with data/numbers on scarcity/droughts from SDGs.

Authors Comment: The suggestions have been addressed, between Lines 42 – 51, as follows:

Clean water and sanitation for all is the goal of sustainable development number 6, one of 17   Sustainable Development Goals set by the United Nations in 2015. Increasing water stress indicates a substantial use of water resources, with greater impacts on the sustainability of these resources, and an increasing potential for conflict among its users. More than 2 billion people live in countries experiencing water stress. Recent estimates show that 31 countries experience water stress between 25% (which is defined as the minimum level of water stress) and 70%. Other 22 countries are above the 70% level and, therefore, are in a severe state of water stress. Water use has increased worldwide by about 1% per year since the 1980s; world demand for water is expected to continue increasing at a similar rate until 2050, which is an increase between 20% and 30% % relative to the current level of use of water [3].

4) Reviewer 1 comment: It is preferred to use functional/characteristic designation of membranes instead of trademark, while trademark and supplier can be specified in brackets.

Authors Comment: The explanation about this condition was inserted in text, between Lines 131 – 133, as follows:

Given the above, the objective of this study was to evaluate the efficiency of two compact systems of rainwater harvesting treatment. One is a filter composed of gravel, sand and activated carbon and the other is a membrane filter using needle punched non woven polyester geotextile membrane.

5) Reviewer 1 comment: Line 76-77: This statement is absolute for certain membrane types.

Authors Comment: The text was changed.

6) Reviewer 1 comment: Line 117: The statement regarding low costs in not supported in the place where it appears.

Authors Comment: The text was withdrawn because it was not the purpose of the work to evaluate costs.

7) Reviewer 1 comment: The city marker is in the sea.

Authors Comment: The figure 1 was altered to show correct location.

8) Reviewer 1 comment: Membrane characteristics are missing.

Authors Comment: The characteristics have been addressed, between lines 205 – 209, as follows:

In this study the efficiency of the membrane filter system was evaluated. Membranes of the type needle punched non woven polyester geotextile (Bidim RT31) were used in the rainwater harvesting treatment process. The membrane used has the following hydraulic properties: permeability normal to plane, kn = 0.37 cm/s; permittivity, y = 0.8 s-1; apparent opening size, O95 = 0.125mm.

9) Reviewer 1 comment: Table 2, 3: Characteristics of filter media should be moved to Materials & Methods.

Authors Comment: The text was moved and included information about granular material measuring in Materials & Methods between lines 171 – 200, as follows:

Sand, gravel and rolled pebbles were washed with water and dried in an oven (Lucadema N 1040) at 100°C for a period of 24 hours before filter construction to reduce initial turbidity and the filter to reach maturation faster. The activated carbon was regenerated by means of a thermal regeneration process; it was activated in an oven at 300°C for 24 hours.

Figure 3. Materials used in the filter - a) Rolled pebbles; b) Gravel; c) Sand; d) Granular activated carbon (GAC).

In order to characterise the filtration materials, tests were carried out to determine the granulometry, void index, specific mass, pH, volatile matter content, ash content, moisture content and bulk density of the sand, gravel and activated carbon, and the iodine number of the activated carbon. All granulometry tests were carried out in the Soil Mechanics Laboratory at UTFPR – Campus Ecoville. Table 1 shows the parameters and the reference methodologies used to characterise the filter materials.

Table 1. Parameters and the reference methodologies used to characterise the filter materials.

Parameter

Sand

Gravel

GAC

Methodology

Unit

Granulometry

X

X

X

ABNT NBR NM 248 (2001)

-

pH

X

X

X

ASTM D 3838-80 (1999)

-

Moisture content

X

X

X

ASTM D 2867-04 (2004)

%

Volatile matter content

X

X

X

ASTM D 5832-98 (2003)

%

Ash content

X

X

X

ASTM D 2866-94 (1999)

%

Specific mass

X

X

X

ABNT NBR NM 23 (2001)

g/cm3

Bulk density

X

X

X

ABNT NBR 52 (2009) Sand ABNT NBR 53 (2009) Gravel

ABNT NBR 12076   (1991) CAG

g/cm3

Iodine number

X

ABNT NBR 12073 (1991)

mg/g

Void index

X

X

X

ABNT NBR 45 (2006)

%

A summary for the results for the characterisation of the filtration materials (sand, gravel, and granular activated carbon) is shown in Table 2.

Table 2. Average values and standard deviation of the parameters used for the physico-chemical characterisation of the sand, gravel, and activated carbon.

Parameter

Sand

Gravel

Activated Carbon

pH

6.8 ± 0.1

8.8 ± 0.1

6.7 ± 0.1

Volatile matter content (%)

1.0   ± 0.1

2.9   ± 0.1

50.8   ± 0.1

Moisture   content (%)

2.38 ± 0.01

0.05 ± 0.01

48.73 ± 0.01

Ash   content (%)

1.11   ± 0.01

6.96   ± 0.01

6.25   ± 0.01

Specific   mass (g/cm3)

2.61 ± 0.01

2.69 ± 0.01

1.27 ± 0.01

Bulk   density (g/cm3)

1.47   ± 0.02

1.38   ± 0.02

0.63   ± 0.02

Void   Index (%)

43.8 ± 0.1

48.8 ± 0.1

32.9 ± 0.1

Iodine   number (mg/g)

-

-

665.86   ± 0.01

The filtration materials used in the filter had degrees of uniformity of 4.9 for sand and 1.9 for gravel, which classifies them as highly uniform.

In Table 3, the granulometric data for the sand and gravel, obtained through granulometry tests, are provided.

Table 3. Granulometric data for sand and gravel.

Material

Minimum Diameter

(mm)

Maximum   Diameter

(mm)

D10

(mm)

D60

(mm)

Sand

0.15

4.76

0.27

1.32

4.90

Gravel

4.76

19.10

7.49

14.60

1.90

 10) Reviewer 1 comment: Figure 5: “in nature” is not a clear designation. Recommended: raw rainwater, after […] treatment. Later, in the Table 4 another designation is used “crude rainwater”. Authors must ensure consistence of terminology.

Authors Comment: Untreated Rainwater altered both designation “in nature” and “crude rainwater”.

11) Reviewer 1 comment: Figure 5: it is not clear if treatment effects are significant and selected treatment is needed, focusing on selected quality parameters.

Authors Comments: The following text has been added in line 399 - 402:

As the rainwater in the evaluated region already shows good quality, the results showed that the filters had similar efficiency between them, but it was not a significant efficiency. However, in regions of low rainwater quality, filters can be more efficient.

Editor comment: It is suggested the revision of English by a native speaker. Some of them have been noted in the attached file.

Authors Comment: The manuscript has been fully revised by a native English speaker.

Reviewer 2 Report

1.  Figure 1 is not clear.  The red point is far from the land of Brazil.  What does it mean?

2.  In Fig. 3, the operation stages and the main materials used in the making of the filter are described.  However, the reviewer cannot find where the membrane is.

3.  The membrane is an absolute filter.  The turbidity of the treated water is less than 0.5.  Since the turbidity of the treated water reached 2.1 shown in Table 4, it seems that the membrane with poor quality is used.  Please give more detailed descriptions of the membrane used in this study.

4.  There are too many discusses in the section Conclusion.

Author Response

Reviewer #2

1) Reviewer 2 comment:   Figure 1 is not clear.  The red point is far from the land of Brazil.  What does it mean?

Authors Comment: The figure 1 was altered to show correct location.

2) Reviewer 2 comment:   The membrane is an absolute filter.  The turbidity of the treated water is less than 0.5.  Since the turbidity of the treated water reached 2.1 shown in Table 4, it seems that the membrane with poor quality is used.  Please give more detailed descriptions of the membrane used in this study.

Authors Comment: The characteristics have been addressed, between lines 205 – 209, as follows:

In this study the efficiency of the membrane filter system was evaluated. Membranes of the type needle punched non woven polyester geotextile (Bidim RT31) were used in the rainwater harvesting treatment process. The membrane used has the following hydraulic properties: permeability normal to plane, kn = 0.37 cm/s; permittivity, y = 0.8 s-1; apparent opening size, O95 = 0.125mm.

3) Reviewer 2 comment:    In Fig. 3, the operation stages and the main materials used in the making of the filter are described.  However, the reviewer cannot find where the membrane is.

Authors Comments: Figure 3 has been improved to show in more detail the location of the membrane.

4) Reviewer 2 comment: There are too many discusses in the section Conclusion.

Authors Comments: The conclusions were better written and the firsts paragraphs were withdraw.

Editor comment: It is suggested the revision of English by a native speaker. Some of them have been noted in the attached file.

Authors Comment: The manuscript has been fully revised by a native English speaker.

Reviewer 3 Report

Please see the attached annotated pdf file

Author Response

Reviewer #3

1) Reviewer 3 comment: Wrong place: this period should be moved at the end of the introduction, after the general discussion on the topic. Moreover, avoid to speak about a specific type of membrane not previously described. Please be more generic

Authors Comment: The place was moved at the end of the introduction and the explanation about this condition was inserted in text, between Lines 131 – 133, as follows:

Given the above, the objective of this study was to evaluate the efficiency of two compact systems of rainwater harvesting treatment. One is a filter composed of gravel, sand and activated carbon and the other is a membrane filter using needle punched non woven polyester geotextile membrane.

2) Reviewer 3 comment:  Maybe the correct term is "rainwater harvesting". Rainwater is always the primary source  for water supply, because river, lakes and groundwater are the various forms under which rainwater is accumulated by natural processes

Authors Comment: The designation was changed.

3) Reviewer 3 comment:  This phrase partially repeats concepts expressed in the previous one. Please merge the two phrases avoiding repetitions – Lines 41 - 43

Authors Comment: The text was withdrawn

4) Reviewer 3 comment:  Please add references supporting these statements – Lines 76 -84

Authors Comment: The following reference has been added:         

[8]

Coelho E. R. C., Di Bernardo. L.,   “Remoção de atrazina e metabólitos pela filtração lenta com leito de areia e   carvão ativado granular.,” Engenharia Sanitária e Ambiental, vol. 17,   pp. 269-276, 2012.

5) Reviewer 3 comment:  Change word comprised

Authors Comment: The word was changed by composed.

6) Reviewer 3 comment:  The sequence in the filter sounds strange to me. Maybe the flow direction is inverted into the filter, because the grain size should decrease downflow, not increasing as represented.

Authors Comments:  In Figure 2 the location of each layer of the granular material was corrected.

7) Reviewer 3 comment:  Figures 2 and 3: What does this question point mean?

Authors Comments:  When the PDF file was generated, the symbol of the pipe diameter in the figure was changed. The correction was made.

8) Reviewer 3 comment: The denomination crude.

Authors Comment: Untreated Rainwater altered both designation “in nature” and “crude rainwater”.

9) Reviewer 3 comment: No information are given in the method section about how did you measured (granular materials) these values. Please give the requested information. And please see previous comment about the lack of methodological information.

Authors Comment: The text was moved and included information about granular material measuring in Materials & Methods between lines 171 - 200

Sand, gravel and rolled pebbles were washed with water and dried in an oven (Lucadema N 1040) at 100°C for a period of 24 hours before filter construction to reduce initial turbidity and the filter to reach maturation faster. The activated carbon was regenerated by means of a thermal regeneration process; it was activated in an oven at 300°C for 24 hours.

Figure 3. Materials used in the filter - a) Rolled pebbles; b) Gravel; c) Sand; d) Granular activated carbon (GAC).

In order to characterise the filtration materials, tests were carried out to determine the granulometry, void index, specific mass, pH, volatile matter content, ash content, moisture content and bulk density of the sand, gravel and activated carbon, and the iodine number of the activated carbon. All granulometry tests were carried out in the Soil Mechanics Laboratory at UTFPR – Campus Ecoville. Table 1 shows the parameters and the reference methodologies used to characterise the filter materials.

Table 1. Parameters and the reference methodologies used to characterise the filter materials.

Parameter

Sand

Gravel

GAC

Methodology

Unit

Granulometry

X

X

X

ABNT NBR NM 248 (2001)

-

pH

X

X

X

ASTM D 3838-80 (1999)

-

Moisture content

X

X

X

ASTM D 2867-04 (2004)

%

Volatile matter content

X

X

X

ASTM D 5832-98 (2003)

%

Ash content

X

X

X

ASTM D 2866-94 (1999)

%

Specific mass

X

X

X

ABNT NBR NM 23 (2001)

g/cm3

Bulk density

X

X

X

ABNT NBR 52 (2009) Sand ABNT NBR 53 (2009) Gravel

ABNT NBR 12076   (1991) CAG

g/cm3

Iodine number

X

ABNT NBR 12073 (1991)

mg/g

Void index

X

X

X

ABNT NBR 45 (2006)

%

A summary for the results for the characterisation of the filtration materials (sand, gravel, and granular activated carbon) is shown in Table 2.

Table 2. Average values and standard deviation of the parameters used for the physico-chemical characterisation of the sand, gravel, and activated carbon.

Parameter

Sand

Gravel

Activated Carbon

pH

6.8 ± 0.1

8.8 ± 0.1

6.7 ± 0.1

Volatile matter content (%)

1.0   ± 0.1

2.9   ± 0.1

50.8   ± 0.1

Moisture   content (%)

2.38 ± 0.01

0.05 ± 0.01

48.73 ± 0.01

Ash   content (%)

1.11   ± 0.01

6.96   ± 0.01

6.25   ± 0.01

Specific   mass (g/cm3)

2.61 ± 0.01

2.69 ± 0.01

1.27 ± 0.01

Bulk   density (g/cm3)

1.47   ± 0.02

1.38   ± 0.02

0.63   ± 0.02

Void   Index (%)

43.8 ± 0.1

48.8 ± 0.1

32.9 ± 0.1

Iodine   number (mg/g)

-

-

665.86   ± 0.01

The filtration materials used in the filter had degrees of uniformity of 4.9 for sand and 1.9 for gravel, which classifies them as highly uniform.

In Table 3, the granulometric data for the sand and gravel, obtained through granulometry tests, are provided.

Table 3. Granulometric data for sand and gravel.

Material

Minimum   Diameter

(mm)

Maximum   Diameter

(mm)

D10

(mm)

D60

(mm)

Sand

0.15

4.76

0.27

1.32

4.90

Gravel

4.76

19.10

7.49

14.60

1.90

10) Reviewer 3 comment:  Lines 239 – 240 - Already written, please do not repeat

Authors Comment: The text was withdrawn

11) Reviewer 3 comment:  Measuring water temperature is nonsensical, because it is controlled by the external conditions at the time of the measurement and it can not give any useful information at all

Authors Comment: The explanation about this condition was inserted in text, between Lines 240 – 245, as follows:

The parameters determined were pH, temperature, turbidity, alkalinity and calcium hardness along with ammonia, nitrite, nitrate and phosphate concentrations. Although the measurement of water temperature is controlled by external environmental conditions, this parameter especially affects pH, so it was considered. The parameters temperature, alkalinity and calcium hardness, although not specified in any Brazilian resolution or standard, were used to compare the efficiency of the treatment and monitoring.

12) Reviewer 3 comment: Membrane characteristics are missing.

Authors Comment: The characteristics have been addressed, between lines 205 – 209, as follows:

In this study the efficiency of the membrane filter system was evaluated. Membranes of the type needle punched non woven polyester geotextile (Bidim RT31) were used in the rainwater harvesting treatment process. The membrane used has the following hydraulic properties: permeability normal to plane, kn = 0.37 cm/s; permittivity, y = 0.8 s-1; apparent opening size, O95 = 0.125mm.

11) Reviewer 1 comment: Figure 5: it is not clear if treatment effects are significant and selected treatment is needed, focusing on selected quality parameters.

Authors Comments: The following text has been added in line 399 -402:

As the rainwater in the evaluated region already shows good quality, the results showed that the filters had similar efficiency between them, but it was not a significant efficiency. However, in regions of low rainwater quality, filters can be more efficient.

12) Reviewer 3 comment: Modification request in Figure 5 (a) and (d)

Authors Comments:  The box-plot graphics are constructed using the median, minimum value, maximum value, 1st and 3rd interquartile. There was an error in the description of the figure. It was mentioned average and not median. The correction was made.

In the text, line 301 - 302, was written as follows:

Box plot graphs containing median, minimum, maximum, 1st and 3rd quarters and interquartile range of the parameters evaluated are showed in Fig. 5.

13) Reviewer 3 comment: In disagreement with the efficiency of the filter for the nitrite parameter.

Authors Comment: The explanation about this condition was inserted in text, between Lines 346 – 350, as follows:

In the case of nitrate, only the highest value obtained for the untreated rainwater was above the limit of the MS Directive 2914/2011 [20] and the CONAMA Resolution 357/2005 [18]. For the ammonia parameters, both filters were efficient in the removal of this contaminant, but not for nitrite, for which the filtration is not influential. Acceptable levels for ammonia, nitrite and nitrate are not specified in the US EPA [19] or the NBR 15527:2007 [21].

14) Reviewer 3 comment: Please look at my previous comments about differences betwen graph and table and justify how do you obtain these percentages of reduction

Authors Comment: There is no mismatch between table and chart data. The table shows the average of each of the parameters and in the box-plot graph the median (a characteristic of the box-plot graph).

15) Reviewer 3 comment: Here you repeat information alredy given. Please modify extrapolating statements of more general interest, as it should be done in a conclusive chapter

Authors Comments: The conclusions were better written and the firsts paragraphs were withdraw.

Editor comment: It is suggested the revision of English by a native speaker. Some of them have been noted in the attached file.

Authors Comment: The manuscript has been fully revised by a native English speaker.

Round 2

Reviewer 1 Report

Title wording can be adjusted: "Comparative analysis of granular and membrane filters for rainwater treatment"

Respond to Comment 2 is not enough. The offered text neither justifies, nor explains what was questioned.

Author Response

General comments

Reviewer #1

1) Reviewer 1 comment: Title wording can be adjusted: "Comparative analysis of granular and membrane filters for rainwater treatment"

Authors Comment: The title was changed according to the reviewer's suggestion.

2.1) Reviewer 1 comment 1:  Respond to Comment 2 is not enough. The offered text neither justifies, nor explains what was questioned.

2.1) Reviewer 1 comment 2: Justification of selection of 8 parameters and if it is sufficient for concluding on non-potable reuse.

Authors Comment: The explanation regarding the Justification of selection of 8 parameters was inserted in text, lines 240 – 243, as follows:

These parameters were used to compare the efficiency of the treatment of the filters and compose a database on the quality of rainwater. In addition, the limits of each one according to standards and guidelines for water quality were assessed.

In addition: The explanation regarding the sufficiency of parameters for concluding on non-potable reuse was addressed in lines 279 – 283 as follows:

According to NBR 15527:2007 [21], rainwater must go through a disinfection process, which may be chlorination, ultraviolet rays, ozone or other processes, defined at the discretion of the designer. The parameters measured in this research are sufficient to guarantee a non-potable use considering that after filtration rainwater must go through a disinfection process.

And: And in the conclusion section, lines 415 – 417, we included:

In addition, even with good quality post-treatment rainwater, the use of such water for non-potable purposes must undergo a disinfection process for the safety of users.

Reviewer 3 Report

After reading your revised manuscript and your replies to my comments I found it suitable for publication. My only comment is the improvement of Fig.1, which is very poor of information: no scale, no coordinates, no geographical names.

Author Response

General comments

Reviewer #3

1) Reviewer 3 comment: My only comment is the improvement of Fig.1, which is very poor of information: no scale, no coordinates, no geographical names.

Authors Comment: Fig. 1 was corrected to include such information.